# Epigenetic Regulation of Breast Cancer Stem Cells Contributing to Carcinogenesis and Therapeutic Implications

**DOI:** 10.3390/ijms22158113

**Published:** 2021-07-29

**Authors:** Hsing-Ju Wu, Pei-Yi Chu

**Affiliations:** 1Department of Biology, National Changhua University of Education, Changhua 500, Taiwan; hildawu09@gmail.com; 2Research Assistant Center, Show Chwan Memorial Hospital, Changhua 500, Taiwan; 3Department of Medical Research, Chang Bing Show Chwan Memorial Hospital, Lukang Town, Changhua 505, Taiwan; 4School of Medicine, College of Medicine, Fu Jen Catholic University, New Taipei City 242, Taiwan; 5Department of Pathology, Show Chwan Memorial Hospital, Changhua 500, Taiwan; 6Department of Health Food, Chung Chou University of Science and Technology, Changhua 510, Taiwan; 7National Institute of Cancer Research, National Health Research Institutes, Tainan 704, Taiwan

**Keywords:** breast cancer, breast cancer stem cells, epigenetic regulation, therapies

## Abstract

Globally, breast cancer has remained the most commonly diagnosed cancer and the leading cause of cancer death among women. Breast cancer is a highly heterogeneous and phenotypically diverse group of diseases, which require different selection of treatments. Breast cancer stem cells (BCSCs), a small subset of cancer cells with stem cell-like properties, play essential roles in breast cancer progression, recurrence, metastasis, chemoresistance and treatments. Epigenetics is defined as inheritable changes in gene expression without alteration in DNA sequence. Epigenetic regulation includes DNA methylation and demethylation, as well as histone modifications. Aberrant epigenetic regulation results in carcinogenesis. In this review, the mechanism of epigenetic regulation involved in carcinogenesis, therapeutic resistance and metastasis of BCSCs will be discussed, and finally, the therapies targeting these biomarkers will be presented.

## 1. Introduction

Globally, breast cancer has remained the most commonly diagnosed cancer and the leading cause of cancer death among women [1]. In 2021, 281,550 new cases of breast cancer were estimated to be diagnosed in women, and 43,600 deaths were predicted from breast cancer in the USA. Therefore, breast cancer has the second highest cancer-related death rate, and is among the most commonly diagnosed cancers in US women [2].

Breast cancer is a highly heterogeneous and phenotypically diverse group of diseases, which require different selection of treatments [3,4,5]. Accurately being able to distinguish between the various subtypes of breast cancer is vital as they have different prognoses and responses to therapy [6]. Gene expression studies have identified six distinct molecular subtypes for breast cancer, which characterize distinct physiological presentation, gene expression profile, prognosis and clinical outcomes [7,8,9]. These subtypes are classified according to the presence or absence of hormone (estrogen (ER) or progesterone (PR)) receptors (HR+/HR-) and human epidermal growth factor receptor 2 (HER2+/HER2-). Luminal A (HR+/HER2-) represents a slow-growing and less aggressive subtype, while luminal B (HR+/HER2+) seems to be more aggressive than luminal A. HER2-positive (HR-/HER2+) breast cancers, which express excess HER2 and do not express hormone receptors, grow and spread more aggressively than other breast cancers and are correlated with poorer prognosis than ER+ breast cancers. Triple-negative or basal-like (HR-/HER2-) breast cancer, with no expression of ER and PR (ER-, PR-) or HER2 (HER2-), represents the worst prognosis subtype. Normal-like breast cancer (HR+/HER2-) is similar to luminal A disease. Although normal-like breast cancer has a good prognosis, its prognosis is still slightly worse than that of luminal A. Lastly, claudin-low tumors are characterized by low genomic instability, mutational burden and proliferation levels [3,4,10,11].

### 1.1. Breast Cancer Stem Cells (BCSCs)

Cancer stem cells (CSCs) are a subpopulation of tumor cells that are endowed with self-renewal and multi-lineage differentiation capacities and play a crucial role in initiation, tumorigenesis, metastasis, chemoresistance and relapse of tumors [12,13,14]. BCSCs are characterized by the expression of cell surface markers, such as CD24−/low, CD44+ and epithelial cell adhesion molecule (EpCAM+) [15,16]. Other surface markers, such as CD133, CD49f, CD90, nestin, ganglioside GD2, C-X-C chemokine receptor type 4 (CXCR4), C-X-C motif chemokine ligand 1 (CXCL1), hydroxymethylglutaryl-CoA synthase (HMGCS), CD166, CD47, aldehyde dehydrogenase 1 (ALDH1) and ATP-binding cassette super-family G member 2 (ABCG2), have also been identified to be associated with BCSCs [17,18,19]. It is now becoming evident that BCSCs can generate different breast cancer subtypes, which express different surface markers due to limited or aberrant differentiation [20,21,22].

Compared to normal cells, BCSCs initiate the multiple changes in gene expression involved in the invasion–metastasis cascade as a result of several mechanisms, including EMT induction and abnormal miRNA biogenesis [23,24,25]. EMT is a complex process that involves many transcription factors, including but not limited to, TWIST, ZEB1, SNAIL, SLUG, Smad interacting protein 1 (SIP1) and E47, and many signaling pathways, such as Wnt/β-catenin, Notch, Hedgehog (HH), nuclear factor-κB (NF-κB)/Akt and transforming growth factor-β (TGF-β)/Smad pathways [26]. Cells undergoing EMT can acquire stem cell-like properties to become CSCs [27]. Intriguingly, BCSCs with a CD44+/CD24−/low phenotype also possess EMT characteristics, such as low expression of E-cadherin (*CDH1*) and high expression of vimentin, N-cadherin (*CDH2*), fibronectin and EMT inducers (Twist, Snail and Slug) [28,29,30]. Since BCSCs play a critical role in carcinogenesis, proliferation and metastasis of breast cancer, targeting BCSCs represents an attractive therapeutic strategy for breast cancer.

### 1.2. Epigenetic Regulation in Normal Function

It has been proven that epigenetic regulation and non-coding RNAs (ncRNAs) are master gene regulators of EMT and CSCs for invasiveness and metastasis of cancer cells [31,32]. Therefore, deciphering the molecular mechanisms that regulate the CSCs’ self-renewal/differentiation balance is urgently required for developing new treatments [33]. In contrast to genetics, epigenetics is defined as inheritable changes in gene expression without alteration in DNA sequence [34]. DNA winds around histone protein to form larger order structural units, nucleosomes, the basic structural units of chromatin. There are two levels of chromatin organization, “open, euchromatin”, which permits active transcription, or “closed, heterochromatin”, which represses transcription. The homeostasis between euchromatin and heterochromatin is determined by epigenetic regulations, including DNA methylation, post-translational histone modifications and alteration of ncRNA expression [35,36].

#### 1.2.1. DNA Methylation and Demethylation

DNA methylation is the most important epigenetic regulation for mRNA and microRNA (miRNA) expression in mammalian cells to ensure normal development and growth [37]; conversely, it is dysregulated in cancer cells [38,39]. In the process of DNA methylation, it creates a ‘fifth base’ from cytosine, 5-methylcytosine (5mC), mostly occurring in CpG islands (CGIs), which act as regulatory hotspots found upstream of the promoter region [40]. There are three types of proteins for DNA methylation and demethylation, including DNA methyltransferases (DNMTs), ten-eleven translocation (TET) enzymes and methyl-binding domain (MBD) proteins [41,42]. Three DNMTs controlling methyl group transfer or CGI methylation consist of DNMT1, responsible for methylation maintenance, and DNMT3A and DNMT3B, capable of de novo methylation, which play critical physiological roles in mammalian genome stability, cellular proliferation and development and cell fate determination [43,44]. Recently, DNMT2 has been identified as a methyltransferase, but for methylation of tRNA instead [45]. The methylated DNA can be recognized by binding MBD proteins to recruit histone-modifying complexes, such as histone methyltransferases (HMTs), for regulating gene transcription and chromatin remodeling [46,47]. It is estimated that 70% of all CGIs in humans are hypermethylated and are found in heterochromatin, which represses transcription. In contrast, hypomethylated CGIs are located in euchromatin, which activates gene expression [48]. Conversely, demethylation is catalyzed by TET family enzymes, TET1, TET2 and TET3, oxygenase enzymes that convert 5mC to 5-hydroxymethylcytosine (5hmC), 5-formyl cytosine (5fmC) and 5-carboxyl methyl cytosine (5CamC) [49,50,51,52].

#### 1.2.2. Histone Modifications

Covalent post-translational modifications (PTMs) of histone tails, including methylation, acetylation, phosphorylation, ubiquitination and SUMOylation, play a pivotal role in modifying gene expression [53]. In contrast to DNA methylation, associating with gene-silencing, histone methylation, acetylation and phosphorylation can change the secondary structure of DNA and result in either induction or prevention of access by transcription factors to gene promoter regions in order to inhibit or activate transcription [53,54].

Histone methylation plays important roles in gene transcription, DNA replication and repair, chromatin organization and developmental and differentiation processes [55,56,57]. Histone methylation, defined as the transfer of one, two or three methyl groups to lysine or arginine residues of histone proteins, is regulated by HMTs and histone demethylases (HDMs) [58]. Transcription silencing is associated with methylation of histone 3 lysine 9, 20 or 27 (H3K9, H3K20 or H3K27), while methylation of histone 3 lysine 4, 36 or 79 (H3K4, H3K36 or H3K79) is involved in transcription activation [59]. Three families of HMTs have been discovered that are specific for the lysine or arginine residue which they modify: the set domain-containing protein family, the non-set domain protein family and the protein arginine methyltransferases (PRMT1) family [57]. A polycomb repressive complex 2 (PRC2) group protein, Enhancer of zeste homolog 2 (EZH2), methylates H3K27 and is a transcriptional repressor [60]. H3K9 methylation occurring in euchromatin causes mono- and di-methylation (H3K9me1 and H3K9me2) catalyzed mainly by G9a, and in heterochromatin, which requires trimethylation (H3K9me3) mostly catalyzed by Suv39H1 and Suv39H2 and results in transcriptional silencing [55,56]. Furthermore, a novel histone lysine methyltransferase, the set and MYND domain-containing protein 3 (SMYD3), methylates H3K4 [61]. On the other hand, two major families of demethylases have been identified, lysine-specific demethylase 1 (LSD1) and Jumonji domain-containing HDMs (JMJD2, JMJD3/UTX and JARIDs). LSD1 specifically demethylates mono- or di-methylated H3K4 or H3K9 and non-histone proteins, such as p53 and DNMT1, indicating that it plays a vital role in a number of normal biological functions and in carcinogenesis, as described in the following section [62]. Similarly, H3K9me3/me2 demethylation is catalyzed by JMJD2C, also known as histone lysine demethylases 4C (KDM4C) [63]. Additionally, JMJD2C demethylates the second methylated histone substrate, H3K36me3 [64].

Histone acetylation occurs via the modifying enzymes, histone acetyltransferases (HATs) or histone deacetylases (HDACs). An acetyl group is added by HATs to ε-amino groups of lysine residues in the histone N-terminal tails, making euchromatin, which allows transcription factor binding and results in gene activation. Conversely, HDACs catalyze the hydrolytic removal of acetyl groups from histone lysine residues, which compact chromatin into heterochromatin, preventing transcription factor binding to DNA and subsequent gene expression [65].

In this review, the mechanism of epigenetic regulation involved in carcinogenesis, therapeutic resistance and metastasis of BCSCs will be discussed, and finally, the therapies targeting these biomarkers will be presented.

## 2. Epigenetic Regulation in Breast Cancer and BCSCs

Approximately 30% of breast cancer is associated with epigenetic modifications, especially DNA methylation [66]. Recently, epigenetic deregulation, such as aberrant DNA methylation and histone modification, has been increasingly recognized to be associated with aberrant gene expression and breast cancer tumorigenesis and metastasis [67,68]. The mechanism of carcinogenesis involves hypermethylation of tumor-suppressor genes and hypomethylation of oncogenes [66]. In addition to DNA methylation, alterations of miRNA expressions have frequently been identified in breast cancer, indicating that miRNAs also play critical roles in the development of breast cancer [51,69,70]. There is an enormous number of epigenetic mechanisms that have been discovered in breast cancer, and only the recent discoveries are elucidated here.

In the formation, maintenance and carcinogenesis of BCSCs, epigenetic modifications including DNA methylation, histone modifications and ncRNA modulation play important roles. There is a large amount of data on HMTs and demethylases; however, there is only a limited amount of data on DNMTs and demethylases in the process of BCSC programming [71,72,73]. Recently, ncRNAs have gained great attention as there are emerging and massive discoveries of ncRNAs involved in BCSC formation. In addition to the contribution to cell self-renewal and differentiation of BCSCs, these epigenetic regulations may distinguish BCSCs from embryonic stem cells (ESCs) and other normal stem cells [74]. Like the normal stem cells, BCSCs have specific DNA methylation signatures distinguishing them from their non-BCSC counterparts [75]. Therefore, all these epigenetic players interact with each other, with chromatin and with transcription factors. Epigenetic regulation in BCSCs involves a complex interplay between proteins and ncRNAs, including miRNAs and long noncoding RNAs (lncRNAs), which are described in more details in the following sections (Table 1). These biomarkers could be classified into tumor activators and tumor suppressors. The distribution of tumor activators and suppressors recently identified in epigenetic regulation of carcinogenesis for BCSC is represented in Figure 1. It is clear to see that most protein and lncRNA biomarkers are tumor activators, whereas most miRNAs are tumor suppressors.

### 2.1. DNMT1

A number of studies have revealed that DNMT1 promotes EMT in TNBC via four main mechanisms [142]. Firstly, EZH2 recruits DNMT1 to form an EZH2-H3K27me3-DNMT1 complex in order to hyper-methylate the promoter of *wwc1*, the EMT suppressor gene, and inhibits its expression in TNBC MDA-MB-231 cells, and the subsequent migration of TNBC cells occurs [143]. Secondly, E-cadherin is suppressed by transcriptional repressors delta-crystallin enhancer binding factor 1 (δEF1) and SIP1. It was demonstrated that the E-cadherin promoter region was hypermethylated, and synergistic inhibition of δEF1 and SIP1 by decitabine treatment de-repressed the E-cadherin expression [144]. Thirdly, DNMT1 is involved in the tumor microenvironment by inducing the oncogenic IL-6/STAT3/NF-κB pathway and promoting the expression of the RNA binding protein, AUF1, in cancer-associated fibroblasts (CAFs) [145]. Lastly, the expression of DNMT1 is suppressed by miR-152 [146] and miR-340 [147], which inhibits the migration of TNBC cells.

Moreover, DNMT1 is indispensable for ESC maintenance, and it was also found to be highly expressed in BCSCs of MMTV-Neu mice, and the DNMT1 deletion protected mice (Dnmt1^fl/fl^-MMTV-Neu-Tg mice) from mammary tumorigenesis by suppressing BCSCs. Additionally, genome-scale methylation studies identified that the gene of insulin gene enhancer protein, *ISL1*, was hypermethylated and downregulated by DNMT1 in breast cancer and BCSCs, and inhibition of DNMT1 or ISL1 overexpression in triple-negative breast cancer (TNBC) cells (CAL51) suppressed CSC populations [78]. It was further revealed that lower *ISL1* transcript expression was significantly correlated with poorer survival in breast cancer patients (*p* < 0.05); however, it was not demonstrated whether *DNMT1* expression was inversely correlated in breast cancer patients [142].

Snail, as a transcription factor and EMT inducer, was also shown to interact with DNMT1, DNMT3A and DNMT3B, and mediated DNA methylation for the promoter of E-cadherin. Clearly, inhibiting Snail-mediated epigenetic regulation resulted in re-expression of E-cadherin and the reversal of EMT. This indicates that Snail is responsible for recruiting these epigenetic enzymes to the promoter of E-cadherin, and thus it is involved in the EMT process [148].

### 2.2. TET1

TET1 was proven to be overexpressed in TNBC patients and associated with a worse overall survival. It contributes to aberrant hypomethylation by TET1 and activation of cancer-specific oncogenic pathways, including PI3K, EGFR and PDGF [149]. The prominent demethylation at the enhancers of ERα, FOCA1 and GATA, well-studied transcriptional factors, has been detected in ER+, luminal A and luminal B types, compared to normal breast tissue [150]. Consistently, it was demonstrated that HDM retinoblastoma-binding protein 2 (RBP2) at the enhancers of ERα may cause tamoxifen resistance in ER+ breast cancers [151]. Hypomethylation in promoters of oncogenes, such as ADAM12, TIMP-1 and the lncRNA HUMT, resulted in elevated expression in TNBC and was associated with poor prognoses [152,153,154]. In contrast, when promoters of tumor suppressor genes are abnormally hypomethylated, they lead to tumorigenesis and invasiveness in TNBCs [142].

Interestingly, a novel H_2_O_2_-regulated pathway linking obesity and BCSCs was recently identified. The TET1, TAR DNA-binding protein (TARDBP), arginine-rich splicing factor 2 (SRSF2) and NANOG genes are all overexpressed in TNBC, especially in TNBC from obese women. Catalase is downregulated in TNBC, which upregulates redox signaling by H_2_O_2_, driving a gene expression cascade from TET1 through TARDBP and SRSF2. Subsequently, methyl-CpG-binding domain protein 2, variant 2 (MBD2_v2), is activated, and finally maintains CSC self-renewal. Furthermore, obesity increases levels of pro-inflammatory signaling factors, such as cytokines, which increase the H_2_O_2_ level in breast cancer cells [155].

### 2.3. HMTs

Deregulation of histone methylation represents another kind of epigenetic event associated with breast cancer invasiveness [25]. For example, G9a specifically methylates p53, a tumor suppressor, at Lys373, and inactivates it. Thus, it has been shown to promote cancer aggressiveness, and its overexpression was correlated with poor prognosis [156,157]. PRMT1 has been demonstrated to promote the EMT program of breast cancer cells by activating ZEB1 [158] and to confer resistance to cetuximab in TNBC cell lines [159], and its overexpression was correlated with cancer malignancy and poor prognosis by methylating and inactivating C/EBPα [160].

#### 2.3.1. Polycomb Group (PcG) Protein

The PcG proteins are transcriptional repressors that act through histone modifications and regulate many developmental and physiological processes, such as cell differentiation and stem cell self-renewal [161]. PcG of proteins, such as BMI1 (B lymphoma Mo-MLV insertion region homolog), a component of the PRC1, and EZH2 (PRC2), are upregulated in breast cancer and BCSCs [80,125,162]. BMI1 is involved in epigenetic regulation of cancer cell proliferation, metastasis, CSC and chemoresistance [80,163]. It was demonstrated that the BMI1 upregulation in 5-fluoro uracil-resistant breast cancer cell lines, such as MDA-MB-231 (mesenchymal stem-like TNBC) and MDA-MB-453 (TNBC) [164]. BMI1 overexpression increased breast cancer sphere formation and promoted EMT, with increased expression of stemness-related genes through activation of Nanog expression via the NF-κB pathway [81]. Recently, downregulation of BMI1 in mouse BCSC line FMMC 419II by the inhibitor, PTC 209, and a stable transfection with a BMI1 shRNA plasmid, correlated with reduced mammosphere formation and a decrease in tumor mass. These results indicate that the inhibition of BMI1 expression in BCSCs might eliminate tumors and relapse [82].

Likewise, increased EZH2 levels have been shown in BCSCs, confirming the role of EZH2 in maintenance of the CSC population [83]. Furthermore, EZH2 was demonstrated to inhibit several tumor suppressor genes, such as E-cadherin, P16 INK4a, BRCA1 and the adrenergic receptor β2 [84]. Conversely, EZH2 was proven to activate NOTCH, resulting in expansion of BCSCs in TNBC [85], consistent with the previous reports that have shown that Notch signaling plays a pivotal role in maintaining the BCSC population [165,166]. In addition, histone modifiers, such as BMI-1 and EZH2, and ncRNAs, such as let7, miR-93, miR-100 and Homeobox transcript antisense RNA (HOTAIR), have all been shown to play roles in the regulation of CSC phenotypes [167].

#### 2.3.2. SETDB1

SETDB1, a HMT, catalyzes the di- and tri-methylation of histone H3K9 to induce gene-silencing [86,87]. Knockdown of SETDB1 results in downregulation of breast cancer formation, migration and invasion, and alteration of EMT/MET makers. The regulatory mechanism involves SMAD7, whose expression is regulated by SETDB1, as SETDB1 knockdown upregulated SMAD7 and suppressed metastasis of breast cancer cells [88]. Moreover, SETDB1 contributes to the EMT process by interacting with the SMAD-7/TGF-β regulatory pathway, which influences SNAI1 (Snail1), an EMT-inducing transcription factor that is associated with metastasis [168,169,170]. However, the interaction of SETDB1 and EMT-inducing transcription factors in BCSC requires further investigation.

### 2.4. LSD1

LSD1 is overexpressed in several types of cancers, including basal-like breast cancer, and is linked to poor prognosis and aggressiveness [171]. The stemness properties of breast cancer proportionately increase with the LSD1 expression [89]. Furthermore, LSD1 induced gene expression involved in EMT and elicited the BCSC program. Phosphorylation of LSD1 at serine-111 (LSD1-s111p) by chromatin anchored protein kinase C-θ (PKC-θ), activated its demethylase activity and promoted EMT [90]. Additionally, miR-708 directly targeted and downregulated the encoding gene of the important epigenetic regulator LSD1. Overexpression of miR-708 in breast cancer cell line MDA-MB-231 inhibits cell proliferation and invasion, while *LSD1* overexpression enhances these processes [172]. In addition, LSD1 is involved in the CXCR4-LASP1 axis, which plays an important role in breast cancer metastasis. CXCR4 signaling raises the nuclear levels of A20 and LSD1. Nuclear-shuttled LASP1 and upregulated LSD1 levels may physically shield Snail1 and prevent access of GSK-3β to Snail, which then represses E-cadherin [173]. Therefore, targeting LSD1 may offer a novel therapeutic strategy to inhibit breast cancer progression and dissemination.

### 2.5. HATs

Likewise, breast cancer development and progression can be activated or inhibited by HATs. Therefore, hyper-acetylation of oncogenes results in cancer progress [72]. MYST3, HAT, was overexpressed in ER+ breast cancer, associated with worse clinical outcomes. MYST3 may activate ERα gene expression by direct binding to ERα promoter, promoting histone acetylation at this locus and altering the chromatin structure. Thus, it was revealed that MYST3 plays a significant role in ER+ breast cancer development, indicating that MYST3 may be a novel target for ER+ breast cancer [174].

### 2.6. HDACs

HDACs also play critical roles in tumorigenesis, including epigenetic regulation of numerous genes for tumor initiation, progression, angiogenesis and metastasis [175]. Histone acetylation also plays a significant role in the epigenetic regulation of CSC miRNAs. MiR-34a, an important tumor suppressor, is suppressed in BCSCs, whereas activation of miR-34a is able to suppress the tumorigenic activity of CSCs. Deacetylation of HSP70 K246 by HDAC1 and HDAC7 enhances cancer cell survival and drug resistance. However, HDAC1 and HDAC7 are targeted and suppressed by miR-34a. Thus, the levels of HDAC1 and HDAC7 are correlated with tumor characteristics, such as grade and stage [91]. More recently, it was demonstrated that CUL4B-Ring E3 ligase (CRL4B) complex interacts with a number of HDAC-containing corepressor complexes, such as MTA1/NuRD, SIN3A, CoREST and NcoR/SMRT complexes. CRL4B/NuRD (MTA1) complexes could be recruited by transcription factors including Snail and ZEB2 and occupy the E-cadherin and AXIN2 promoters to induce tumorigenesis and breast cancer cell invasion [176].

### 2.7. ncRNAs

ncRNAs are emerging as important regulators in gene expression and tumorigenesis [177,178]. There are two categories of ncRNAs according to the size and structural or regulatory characteristics. lncRNAs are ncRNAs > 200 nucleotides, while ncRNAs < 200 nt include miRNAs, small nucleolar RNAs (snoRNAs) and piwi RNAs (piRNAs) [178]. ncRNAs play important roles in regulating gene expression by interacting with epigenetic modifiers, and their dysregulation appears to associate with epigenetic alterations in cancer [179] (Table 1).

#### 2.7.1. miRNAs

The investigations on miRNAs are emerging and prevalent. miRNAs are short, single-stranded ncRNAs (18~22 nucleotides in length) that negatively regulate mRNAs post-translationally by binding to the 3′-UTR region of mRNA [180,181]. There is a growing list of miRNA genes abnormally methylated in cancer, resulting in dysfunction in normal biological processes and carcinogenesis [179]. There are two categories for cancer-associated miRNAs: the oncogenic miRNAs (oncomiRs) are usually highly expressed, and they contribute to cancer development and progression and could be useful for diagnosis, prognosis and treatment [69,70,182], and the tumor-suppressive miRNAs (miRsupps), which inhibit tumorigenesis by regulating cell proliferation, apoptosis, invasion, metastasis and therapeutic resistance in BCSCs [25,183] (Figure 2). For example, miR-31 is overexpressed in breast cancer and is directly activated by the NF-κB signaling pathway. Downregulating miR-31 leads to a reduced BCSC subpopulation and diminishes the abilities of tumor initiation and metastasis. Furthermore, miR-31 was shown to mainly upregulate Wnt/β-catenin activity by suppressing Xin1, Gsk3β and Dkk1, and to inhibit TGF-β signaling through Smad3 and Smad4 [99].

Investigation of methylation of five miRNAs associated with the invasiveness and metastasis of breast cancer revealed that miR-132, miR-137 and miR-1258 were hypermethylated and associated with clinical features [69]. Profile analysis of methylation of miRNA and differentially methylated regions (DMRs) showed that miR-31, miR-135b and miR-138-1 were associated with methylation in early and late postpartum groups of breast cancer [184]. Downregulation of miR-205 expression was mediated by Erb-B2 receptor tyrosine kinase 2 (ErbB2) signaling via the Ras/Raf/MEK/ERK pathway and hypermethylation of the miR205 promoters. This led to increased ErbB2 tumorigenicity [185].

In investigating the epigenetic regulation and mechanisms involved in BCSC, NIMA-related kinase 2 (NEK2) has been identified as a novel target of miR-128-3p, and is upregulated in breast cancer. The overexpression of miR-128-3p was demonstrated to inhibit cell proliferation, invasion, migration and self-renewal of BCSCs and resulting tumorigenicity. It was further demonstrated that downregulation of NEK2 promotes the inhibition of the Wnt pathway. Thus, NEK2 represents a novel target for breast cancer treatment [138]. Similarly, miR-600 is a tumor suppressor. Overexpression of miR-600 suppressed BCSC self-renewal by suppressing stearoyl desaturase 1 (SCD1) and subsequently inhibiting Wnt. This leads to decreased in vivo tumorigenicity and good prognosis. Its silencing promotes BCSC expansion, and Wnt signaling activation promotes self-renewal [137].

Interestingly, there is extensive overlap between miRNAs with both the EMT process, an important mechanism for carcinogenesis of BCSCs, and with BCSC phenotype, and most of them are epigenetically regulated, since they are located in or around CGIs [13,186]. Numerous miRNAs contribute to BCSC formation by regulating the EMT process. It was demonstrated that miR-146a, a tumor suppressor, plays a role in regulating the induction and maintenance of BCSCs during EMT, and this identified a novel mechanism for breast cancer development [134,135]. Mechanistically, miR-146 binds and degrades the 3′UTR of LIN28, and LIN8 binds to the let-7 pre-miRNA and blocks production of mature let-7. Therefore, miR-146a upregulated the Let-7c level by degrading LIN28. Furthermore, let-7 controls Wnt signaling pathway activity and could be enhanced by the miR-146 inhibition of H19, resulting in a positive feedback regulation loop with let-7. This miR-146a/Lin28/Wnt signaling axis prevents symmetric cell division and inhibits the BCSC expansion [136]. Furthermore, the reduced expression of let-7 (tumor suppressor) enhances the self-renewal capacity in BCSCs [187]. Additionally, it was demonstrated that let-7c reduces self-renewal of ERα+ BCSCs and decreases ERα expression through directly binding to the 3′UTR, inhibiting estrogen-induced activation of Wnt signaling [129,130].

Previous studies have found that miR-10b and miR-23a were upregulated by TGF-β1 and that overexpression of these two miRNAs resulted in EMT, proliferation, invasion and metastasis of breast cancer. Additionally, miR-23a suppressed the *CDH1* gene, which in turn activated Wnt/β-catenin signaling. These results suggest that both miR-10b and miR-23a promote TGF-β1-induced EMT and tumor metastasis in breast cancer [92,94]. Furthermore, miR-10b induces the metastasis and migration of BCSCs; therefore, overexpression of miR-10b in MCF-7 cells enhanced self-renewal and the expression of stemness and EMT markers. It was further identified that phosphatase and tensin homolog (PTEN), a key regulator of the PI3K/AKT pathway, is a potential target of miR-10b. miR10b suppression upregulated PTEN and downregulated AKT. These data clearly demonstrated that miR-10b upregulates the self-renewal and migration of BCSC by inhibiting the PTEN/PI3K/AKT pathway [93].

In addition, there are several studies investigating the role of individual miR-200 cluster members in EMT. For example, the expression of Kindlin-2, a regulator of integrin functioning, was closely associated with the metastatic phenotype of breast cancer, and was directly targeted and inhibited by miR-200b, leading to the inhibition of EMT and metastasis [128,188]. Recently, it was revealed that p53 upregulates the miR-30a expression by binding to the *MIR30A* promoter in TNBC, and miR-30a suppresses the *ZEB2* expression. Decreased miR-30a expression was associated with p53 deficiency, LNM and poor prognosis. These results indicate that tumor aggressiveness is regulated by the novel p53/miR-30a/ZEB2 axis, subsequently inhibiting the miR-200c expression [131]. Similarly, miR-590-5p and miR-140 acted as tumor suppressors and inhibited breast cancer stemness by targeting the *SOX2* gene, and both led to reducing the BCSC population [132,133]. In contrast to miR-200, miR-221 is an oncomiR, and its overexpression contributes to stemness maintenance in breast cancer by directly targeting *CDH1*, resulting in E-cadherin inhibition. This study revealed a novel mechanism in which E-cadherin is post-transcriptionally inhibited by the Slug-promoted miR-221 overexpression [95]. It was also found that miR-221 was upregulated in BCSCs. miR-221 targets and suppresses the *DNMT3B* gene, leading to extensive changes in the DNA methylation of several promoter regions, such as *NANOG* and *OCT 3/4*, stem cell pluripotency regulators [96]. The miR-221/222 cluster functions as an oncogene since it induces the growth, migration, invasion and propagation of BCSCs. Mechanistically, the tumor suppressor PTEN was demonstrated as a target of miR-221/222, and downregulation of PTEN induces AKT phosphorylation [97]. It was further revealed that ectopic expression of miR221/222 or PTEN knockdown involved the overexpression of the AKT/NF-κB p65/COX-2 pathway in BCSCs [98].

In addition, it has been reported that DNA methylation plays an important role in deregulation of miR-124, miR-125b, miR-203 and miR-375, which are involved in EMT and breast cancer progression [189,190]. Damiano et al. proved that DNA methylation is important for epigenetic regulation of the miR-200c/ZEB1 axis [191], which is consistent with the previous findings that ZEB1 and ZEB2 can inhibit the transcription of the miR-200 cluster and induce EMT and aggressiveness in breast cancer [192,193]. Polyl-isomerase Pin1 was identified as another target of miR-200c involved in BCSC expansion, invasiveness and tumorigenicity [126].

DNMT1 has also been involved in TNBC CSCs through epigenetic regulation. The tumor-suppressor miR-137 had significantly lower expression in TNBC tissues (*p* < 0.05) compared with adjacent normal tissues. miR-137 suppressed *BCL11A* expression by directly targeting its 3′UTR, leading to inhibition of tumorospheres’ BCSC population. BCL11A could interact with DNMT1, and silencing of either BCL11A or DNMT1 suppresses stemness and tumorigenesis of TNBC cells through inhibiting ISL1 [194]. In addition, miR-137 reduces the FSTL1 expression. FSTL1 promotes oncogenesis in breast cancer by inducing stemness and chemoresistance via activating Wnt/β-catenin signaling through integrin β3. These results revealed a miR-137/FSTL1/integrin β3/Wnt/β-catenin signaling axis in regulating stemness and chemoresistance [139].

Upregulated levels of tumor-suppressive miRNAs, miR-200a, miR-200b, miR-15a, miR-429 and miR-203, lead to downregulation of the PRC1 group of proteins, such as BMI1, RING1A, RING1B and Ub-H2A. Notably, increased expression of miR-200a, miR-200b and miR-15 also results in decreased BMI1 and Ub-H2A protein expression in the CD44+ BCSC population of MDA-MB-231 cells. Upregulated levels of BMI1 regulate miRNAs, promoting mesenchymal to epithelial transition (MET), in which N-Cadherin, Vimentin, β-Catenin, Zeb and Snail are regulated, and ultimately leading to inhibition in proliferation, migration and invasion [127]. This indicates the chemo-sensitizing activity of the miRNAs in addition to the tumor-suppressive activity [195].

More recently, it was discovered that the overexpression of PD-L1 promotes chemoresistance and enhances stemness-like properties of BCSCs via activation of PI3K/AKT and ERK1/2 signaling pathways. miR-873 suppressed PD-L1 via inhibiting downstream PI3K/AKT and ERK1/2 signaling, leading to suppressing stemness and chemoresistance of BCSCs [140]. miR-520b, an oncogene, is upregulated in breast cancer tissue and BCSCs and promotes the stemness that predicts poor prognosis in patients. miR-520b upregulates Hippo/YAP signaling via targeting LATS2 to promote breast cancer stemness and maintenance [100].

These results indicate that miRNAs play critical roles in BCSC formation, thus representing promising targets for cancer treatment.

#### 2.7.2. lncRNAs

In addition to miRNAs, lncRNAs (200 nucleotides) have emerged as new players in stem cell signaling through multiple mechanisms, mainly via transcriptional, post-transcriptional and epigenetic regulation of genes and proteins, and both play pivotal roles in tumorigenesis, pluripotency, apoptosis, chemoresistance, angiogenesis, self-renewal and metastasis in BCSC subpopulations [183,196,197]. Unlike miRNAs and mRNAs, lncRNAs have low expression in breast [198].

Dysregulation of lncRNAs was demonstrated to play a pivotal role in tumorigenesis [179]. Interestingly, lncRNAs could act as sponges of miRNAs in BCSC. Hence, lncRNAs could also regulate the expression levels of the targets of miRNA, and this results in a complex interplay among lncRNAs, miRNAs and proteins [183]. MANCR (mitotically associated noncoding RNA) is upregulated in breast cancer, and downregulation of MANCR reduces TNBC cell proliferation [199].

Notably, it has been shown that a higher rate of hypomethylation is observed in many lncRNAs in breast cancer, in contrast to the well-known phenomenon of the CGI hypermethylation phenotype (CIMP) in tumors [150]. The hypomethylation of lncRNA EPIC1 (epigenetically induced lncRNA1) upregulates its expression, and overexpression of EPIC1 is correlated to poor survival outcomes in luminal B breast cancer, and was further revealed to promote tumorigenesis through interacting with MYC to increase the occupation of MYC target genes [116]. Although a lot of efforts have been made to identify oncogenic drivers for breast cancer, the lncRNA sequences and epigenetic factors remain largely unexplored by comparison [6].

Recently, there have been numerous lncRNAs identified to be involved in BCSC formation, and most of them are oncogenic. lncRNA-HAL is elevated in the quiescent stem cell population of MCF-7-MCTS. lncRNA-HAL is associated with cell proliferation, migration and cell survival; subsequently, lncRNA-HAL silencing impairs the proportion and function of BCSCs [114]. It was also demonstrated that lncRNA-LINC01133 stimulates the BCSC phenotype and growth characteristics in TNBC. Additionally, it has been shown that lncRNA-LINC01133 is a direct mediator of the upregulation of the miR-199a-FOXP2 signaling pathway and a critical regulator of Kruppel-Like Factor 4 (KLF4), the pluripotency-determining gene [115]. Similarly, Nuclear lncRNA metastasis-associated lung adenocarcinoma transcript 1 (MALAT1) is elevated in BCSC MCF7, and its knockdown inhibits proliferation, mammosphere formation, invasion and migration of BCSC MCF7. Mechanistically, MALAT-1 regulates Sox-2, the stemness factor [122]. lncRNA FEZF1-AS1 is also upregulated in breast cancer tissue and is correlated with poor prognosis in patients. FEZF1-AS1 downregulation inhibits mammosphere formation, the expression of stem cell markers and the rate of CD44+/CD24− production. Subsequently, cell proliferation, migration and invasion are significantly inhibited. Further mechanistic analysis demonstrated that FEZF1-AS1 modulates BCSC and Nanog expression through sponging miR-30a [123]. Likewise, lncRNA LINC00511 is highly expressed in breast cancer and is associated with poor prognosis of patients. Mechanistically, LINC00511 acts as a miR-185-3p sponge and binds to E2F1, upregulating Nanog. As a result, the LINC00511/miR-185-3p/E2F1/Nanog axis promotes stemness and tumorigenesis of BCSCs [124].

LncCCAT1 (colon cancer-associated transcript-1) is significantly overexpressed in breast cancer tissue and BCSCs, associating with poor patient outcomes. It was mechanistically demonstrated that LncCCAT1 can interact with miR-204/211, miR-148a/152 and Annexin A2, upregulating T-cell factor 4 or translocating β-catenin to the nucleus Annexin A2, leading to activating the Wnt pathway and inducing the proliferation, stemness, invasion and migration of BCSCs [120]. It was also reported that LncRNA LUCAT1 (lung cancer-associated transcript 1) is highly expressed in BCSCs. LUCAT1 activates stemness features in breast cancer through the Wnt/β-catenin signaling pathway. The overexpression of LUCAT1 is correlated with tumor size, LNM, poor prognosis and shorter survival. Additionally, LUCAT1 and the transcription factor TCF7L2 are targets of miR-5582-3p, thus LUCAT1 functions as a miR-5582-3p sponge, promoting the Wnt/β-catenin axis [121]. Finally, lncRNA THOR was elevated in TNBC. Intriguingly, silencing THOR induces decreased mammosphere formation, stemness marker expression and ALDH1 activity of BCSC. Mechanically, THOR binds directly and upregulates β-catenin in order to promote BCSC stemness [119].

Conversely, lncRNA FGF13-AS1 was downregulated in breast cancer. FGF13-AS1 as a tumor suppressor suppresses breast cancer cell proliferation, invasion and migration by impairing glycolysis and stemness characteristics. Through decreasing the half-life of c-Myc mRNA and inhibition of the binding between insulin-like growth factor 2 (IGF2BPs) and c-Myc mRNA, Myc inactivates FGF13-AS1 [141].

Like miRNAs, lncRNAs maintain CSCs properties by inducing EMT and self-renewal signaling pathways [33]. SOX2 is elevated in BCSCs by lncRNAs, such as SOX2 overlapping transcript (SOX2OT) [105] and linc00617 [106]. Moreover, the self-renewal HH-GLI1 signaling pathway is activated by lncRNA-Hh, leading to increased SOX and OCT4 levels and promoting CSCs maintenance. GAS1, an enhancer of HH signaling, is directly targeted by lncRNA-Hh. This increases the Sox2 and Oct4 expression and promotes the acquisition of stemness traits [107]. Furthermore, the lncRNA SOX21-AS1 is upregulated in breast cancer tissue and is correlated with poor prognosis. Notably, SOX21-AS1 knockdown was proven to diminish proliferation, invasion and stem factor, i.e., Nanog, LIN28, Oct4 and SOX2, of BCSCs. This study indicates that SOX21-AS1 regulates properties and carcinogenesis of BCSC via targeting SOX2 [117]. It was further confirmed that the lncRNA SOX21-AS1 is overexpressed in BCSCs of MCF-7 and MDA-MB-231, enhancing stemness of BCSC as well as proliferation, invasion and migration in CSC-MCF-7 cells via suppressing the Hippo signaling pathway [118]. LINC00617 is elevated in breast cancer samples and functions as an important regulator of EMT, enhancing the progression and metastasis via upregulating Sox2 [106].

Despite the enormous findings on lncRNAs, there are many lncRNAs that remain to be identified with further studies to investigate their roles in BCSCs. Like miRNAs, lncRNAs are promising therapeutic targets due to their oncogenic functions.

## 3. Therapeutic Implications

Since epigenetic modifications are reversible, and aberrations in DNA methylation and histone modification are often associated with tumorigenesis, they appear to be potential therapeutic targets for cancer patients [34,72]. The proteins and ncRNAs involved in epigenetic regulation in breast cancer cells and BCSCs described above represent potential and promising therapeutic targets, and the development of therapies are under investigation or in clinical trials (Table 2). These investigations revealed new strategies for personalized medicine [25].

### 3.1. Methylation-Based Therapy

DNMT1 inhibitors (DNMTi) can be classified into two categories, nucleoside analogues and non-nucleoside analogues. Nucleoside analogues incorporate into DNA as cytosine mimics and promote proteasomal degradation for DNMTs, leading to DNA hypomethylation. On the other hand, non-nucleoside analogues do not mimic cytosine for DNA incorporation, but they bind directly and inhibit specific target proteins instead [234,235].

DNMTi, the nucleoside analogues 5-azacytidine (5-AzaC (Vidaza)) and 5-aza-20-deoxycytidine 5-AzaDC (decitabine), have been FDA-approved for treating hematological malignancies, including acute myeloid leukemia (AML) and myelodysplastic syndrome (MDS) [200]. 5-AzaC was demonstrated to lead to hypomethylation and gene de-repression, resulting in preventing EMT in vitro [201]. Experimental data have revealed the antitumor effect of both azacitidine and decitabine against TNBC cells. The protein levels of DNMTs were associated with response to decitabine in chemotherapy-sensitive and -resistant TNBC cells examined in TNBC patient-derived xenograft organoids, and all three DNMTs, DNMT1, DNMT3A and DNMT3B, were degraded by decitabine treatment both in vitro and in vivo. Furthermore, blocking DNMTs’ degradation promoted resistance to decitabine, identifying that DNMT levels in TNBC patient-derived xenograft mouse models are response biomarkers to decitabine [236]. However, their clinical efficacy in breast cancers has remained controversial because the combination therapy involving azacitidine in breast cancer has not yielded satisfactory results [142]. Notably, a phase II clinical trial including the combination of 5-AzaC and HDACi, entinostat, did not show significant clinical efficacies, since none of the 13 TNBC patients achieved partial response and the primary endpoint was not met [237]. Since breast cancer is highly heterogeneous, certain treatments might only benefit specific cancer subgroups, and patients should be stratified for specific therapies.

Guadecitabine (SGI-110), a dinucleotide of decitabine and hypomethylating agent, is a second-generation DNMTi. Guadecitabine is more resistant than azacitidine or guadecitabine to degradation by cytidine-deaminase (CDA), and has greater incorporation into DNA of dividing cancer cells [202]. There are growing evidences revealing anti-tumor immunogenicity of guadecitabine through activating cytotoxic T cells (CTLs) [142]. Myeloid-derived suppressor cells (MDSCs) are increased in TNBC patients compared with non-TNBC patients [238], leading to the inhibition of anti-tumor T cell immunity in the TNBC tumor microenvironment [239,240], and activation of growth, metastasis and CSC populations in TNBC [238]. Recently, it was demonstrated that guadecitabine reduced MDSCs in vivo (4T1 cells in BALB/cJ mice), and decreased the tumor burden via CTL-mediated anti-tumor response [142].

### 3.2. Demethylation-Based Therapy

Since epigenetic changes are reversible, the HDM JMJD2C represents a promising therapeutic target for different cancers [203]. Ye et al. examined the therapeutic efficacy of a novel JMJD2 inhibitor, NCDM-32B. Indeed, NCDM-32B treatment attenuated the cell viability and anchorage-independent growth in breast cancer, and the mechanism involved several vital signaling pathways that activate cell proliferation and transformation [203].

### 3.3. Chromatin Modifier Therapy

Histone lysine methylases are highly promising therapeutic targets due to the specificity of the targets. The main histone lysine methylase inhibitors approved by the FDA target EZH2, G9a/GLP, disruptor of telomeric silencing 1-like (DOT-1L) and SUV39h1 [241]. Despite the promising results, 3-Deazaneplanocin (DZNep), an inhibitor of EZ2H and H3K9me3, has a short plasma half-life, has nonspecific suppression of histone methylation and is toxic in animal models [242]. Therefore, several EZH2 inhibitors have been developed in order to improve antitumor activity and decrease toxicity [242]. ZLD1039, an EZH2 inhibitor, also exhibited a strong anti-cancer effect by suppressing breast cancer growth and metastasis [204].

Ho et al. showed the use of the G9a-specific inhibitor, BIX-01294, to effectively abrogate G9a’s actions in a breast cancer model. This resulted in inducing apoptosis and impairing cell migration, cell cycle and anchorage-dependent growth in breast cancer cells. However, unexpectedly, G9a inhibition also led to the promotion of pro-tumorigenic pathways, such as the hypoxia-induced pathway (HIF), even in normoxic conditions that may facilitate the tumor-suppressive effects of BIX-01294 [205]. This stressed the importance of HMTs worthwhile for further investigation, such as SETDB1, in drug development. Moreover, there are currently no HMTs in clinical trials for treating breast cancer; therefore, this highlights the urgent need for developing drugs targeting HMT [168].

There are a number of LSD1 inhibitors that have been developed and examined for their effects in cancers. These contain bizine, the tranylcypromine derivatives NCL1 and GSK2879552, biguanide and bisguanidine polyamine analogs [209]. GSK287 is an orally bioavailable, irreversible LSD1 inhibitor, currently under clinical evaluation for cancer treatment [210]. The polyamine analog inhibitors of LSD1, 2d or PG11144, alter promoter activity of multiple genes in breast cancer cells and are proposed to have considerable therapeutic modality [211]. Importantly, the LSD1 inhibitor pargyline partially inhibited CSC formation. Furthermore, inhibiting LSD1 suppressed EMT and stemness-like resistance signatures combined with chemotherapy. Combination of LSD1 and chemotherapy suppressed tumor growth in vivo compared to chemotherapy alone [90]. It was also demonstrated that the LSD1 inhibitor, QC6352, suppressed tumor growth, sphere formation and proliferation of patient-derived BCSCs in a newly developed orthotopic xenograft model [212]. In addition, the application of a dual HDM inhibitor (MC3324) in cell culture induces significant growth arrest and apoptosis and shows the same hormone modulatory effects in downregulating ERα as an established hormone therapy. Likewise, MC3324 displays tumor-selective potential in both xenograft mice and chicken embryo models, with no toxicity and good oral efficacy [213]. There are a number of LSD1 inhibitors recently developed, such as iadademstat (ORY-1001) and KDM1A inhibitor NCD38, targeting BCSCs in vitro and in vivo [214,215,227], and they offer therapeutic potential. These findings showed that the LSD1 inhibitor, iadademstat (ORY-100), could inhibit colony formation and SOX2 expression, especially in multi-drug-resistant human luminal B tumors [214].

HDACi enhance cellular protein acetylation by inhibiting HDAC activity. There are five classes of HDACi: hydroxamic acids (pan-HDACIs, SAHA, vorinostat), benzamides (specific class I HDACIs, entinostat (MS-275)), cyclic tetrapeptides (specific class I HDACIs, romidepsin), short-chain fatty acids (class I and II HDACi, valproic acid) and sirtuin inhibitors (pan-HDACIs, including SIRT1 and SIRT2, nicotinamide) [175]. There are a number of pre-clinical in vivo and in vitro studies support the efficacy of HDACi, such as TSA, Romidepsin (FK-288), Vorionstat (SAHA) and Panobinostat, and they sensitize the mesenchymal-like TNBC cell line and are useful in treating advanced breast cancers [195,243]. HDACi such as TSA or SAHA promote PTEN membrane translocation through PTEN acetylation at K163 via inhibiting HDAC6. This K163 acetylation suppressed the interaction of the PTEN C-tail with the remaining part of PTEN, leading to PTEN membrane translocation. This PTEN activation leads to inhibition of cell proliferation, invasion, migration and tumor growth. Hence, non-selective HDAC or HDAC6-specific inhibitors may be more clinically suitable for treating tumors without PTEN mutations or deletions [206].

Targeting BCSCs is a promising therapeutic strategy. For example, Witt et al. [207] have proven that compared with non-stem tumor cells, HDAC1 and HDAC7 are overexpressed in BCSCs. Further, currently available HDACi such as TSA, a pan HDACi, inhibit HDAC1 and HDAC7, that may therefore modify the epigenetic markers that characterize CSCs. On the other hand, Hsieh et al. [208] revealed that HDAC3 was mechanistically linked to BCSC homeostasis by enhancing β-catenin expression through the Akt/glycogen synthase kinase 3β (GSK3β) signaling pathway. They applied a pan-HDACi, AR-42, as a HDAC3-selective inhibitor, and showed high potency and isoform selectivity in inhibiting HDAC3. Importantly, they exhibited in vitro and/or in vivo efficacy in inhibiting the CSC subpopulation of TNBC cells via downregulating β-catenin. Additionally, other preclinical models have demonstrated the efficacy of HDACi against BCSCs [208,244].

### 3.4. miRNA-Based Therapy

As elucidated above, ncRNAs play pivotal roles in carcinogenesis, especially in BCSC initiation, maintenance and proliferation; therefore, applying them as the drug targets poses a big challenge for therapeutic drugs. There are emerging investigations for lncRNAs. However, both experimental and clinical modulation of lncRNA-based treatment are not yet available. There are two strategies for miRNA-based therapy, inhibiting oncomiRs by using miRNA antagonists and restoring miRsupps by miRNA replacement therapy [245]. The main factor for effective miRNA therapy is finding an accurate delivery system to make sure miRNAs reach the specific site.

miR-34 is the miRsupps; thus, epigenetic restoration of miR-34 could sensitize cancer cells to drugs and inhibit BCSCs. The miR-34-based drug MRX34 passed the phase I clinical trial [216]. Furthermore, caffeic acid, a hydroxycinnamic acid (3,4-dihydroxycinnamic acid, CaA), enhanced the expression of miR-148a by inducing DNA methylation. Subsequently, miR-148a suppressed the TGF-β-SMAD2 signaling pathway and the BCSC properties [217]. Similarly, glabridin, a phytochemical from the root of *Glycyrrhiza glabra*, enhanced the expression of miR-148a via DNA demethylation. miR-148a then inhibited SMAD2, and in turn attenuated the BCSC properties [218]. TV-circRGPD6 inhibits BCSC metastasis via the miR-26b/YAF2 axis. Therefore, this represents a novel therapeutic strategy to treat breast cancer [221]. As described above, let-7a targets BCSC via inhibiting ERα and involving let-7c/ER/Wnt signaling [219]. Another study also applied let-7a as a therapeutic agent for targeting BCSCs when they are formulated in Herceptin-conjugated cationic immuno-liposome (mi/siRNA-loaded PEGylated liposome conjugated with Herceptin (Her-PEG-Lipo-mi/siRNA)) with hyaluronic acid and protamine. This efficient liposomal delivery system for the combination of miRNA and siRNA to target BCSC can be exploited as an efficacious therapeutic modality for breast cancer [220].

### 3.5. Combination Therapy

Considering the crosstalk between signaling pathways in breast cancer cells and BCSCs, combination therapy is necessary in order to overcome the drug resistance. Poly(ADP-ribose) polymerase inhibitors (PARPi) are effective anticancer drugs, producing good initial clinical responses, especially in homologous recombination DNA repair-deficient cancers. However, resistance always occurs, possibly due to compensatory pathways, genes and effector proteins overlap. Interestingly, epigenetic medication might synergize with PARPi in several ways [229]. It was reported that DNMTi combined with the PARPi, talazoparib, promoted tight binding of talazoparib to DNA and increased double-strand breaks’ (DSB) formation and cytotoxicity in TNBC stem cell-like cell lines due to increased amounts of PARP1 at locations of DNA damage [223]. Recently, it was discovered that DNMTi increases poly(ADP ribose) polymerase (PARP) trapping and reprograms the DNA damage response to cause homologous recombination deficiency (HRD), sensitizing BRCA-proficient cancer cells to PARPi. Mechanistically, DNMTi combined with PARPi promotes broad innate immune signaling, driven in part by stimulator of interferon genes (STING), to generate HRD. This combination strategy will be tested in a phase I/II TNBC clinical trial [224]. Yamaguchi et al. [225] showed that EZH2 inhibition contributes to PARPi sensitivity in breast cancer cells. Thus, combining PARP and EZH2 inhibition represents a promising therapeutic strategy in breast cancer.

Combinations of epigenetic drugs and immunotherapy are gaining attention. In the combination of HDACi and immune checkpoint inhibitors, HDACi upregulates PD-L1 in TNBC cells. This leads to a significant inhibition in tumor growth and increased survival, associated with elevated T cell tumor infiltration and decreased CD4(+) Foxp3(+) T cells [230]. The combination therapy incorporating epigenetic drugs such as lysine-specific histone demethylase-1A inhibitors or HDACi with PARPi/anti-PD-1/PD-L1 represents a novel, potentially synergistic strategy for targeting BCSCs and overcoming resistance and recurrence [229]. A phase II study has just started to evaluate the clinical efficacy of the combination therapy of vorinostat, pembrolizumab and tamoxifen in ER+ metastatic breast cancer [246].

A recent study revealed that the second-generation DNMTI, guadecitabine, in combination with the HDACi, entinostat, could convert highly aggressive TNBC cells that have undergone EMT into a less aggressive phenotype. Moreover, the combination therapy suppressed TNBC cell proliferation, colony formation, motility and stemness of the cancer cells in vitro, and with anti-tumor effects in patient-derived xenograft mouse models [222].

Additionally, the combination therapy of HDACi YCW1 with ionizing radiation enhanced toxicity and increased autophage induction. This led to induction of cell death in TBNC cell lines in vitro and in mouse models [233]. In the combination of HDACi and JAK/BRD4 inhibitors, JAK/BRD4 inhibition sensitizes TNBC cells to HDAC inhibitions [231]. As a pan-JAK1/2 inhibitor (Ruxolitinib) is already in clinical use for treatment of myelofibrosis [247], this combination therapy could be a promising therapeutic strategy. A phase Ib clinical trial investigated the combination of the chemotherapeutic drug ixabepilon with vorinostat for metastatic breast cancer patients. The objective response rates were comparable to those achieved with the previously approved ixabepilon monotherapy or the combination with capecitabine. Most promising was the fact that there were fewer side effects [232].

There are several studies focused on combining LSD1 inhibitors with other therapies. It was demonstrated that HDACs regulates LSD1 to control breast cancer cell growth. Combined therapy of TNBC cells with the LSD1 inhibitor, pargyline, and the HDACi, SAHA, leads to inhibition of cell growth [226]. It has been shown that LSD1 inhibition reduced CSC potential, as well as sensitizing human breast cancer cells to chemotherapy when combined with chemotherapeutic agents such as doxorubicin [227]. LSD1 inhibition in combining with PD-1 antibodies significantly inhibited tumor growth and pulmonary metastasis in a TNBC mouse model [228].

## 4. Conclusions and Future Directions

There is a growing list of proteins and ncRNAs identified in epigenetic regulation that may represent useful biomarkers for diagnosis and/or prognosis for breast cancer. The major challenges in cancer therapy are tumor recurrence and resistance to conventional therapies, such as chemotherapy and radiotherapy, and CSCs are the major players in these events. Therefore, comprehensive elucidation of regulatory mechanisms in BCSCs will definitely help to develop more effective precision medicine. There is emerging data on dysregulation of ncRNAs, and ncRNA hyper/hypomethylation contributes to cancer stemness. There are currently not many miRNA-based therapies for breast cancer; therefore, these represent a great opportunity in developing novel therapeutic strategies for breast cancer. Additionally, ncRNAs have the advantage of multi-target characteristics, which should minimize the possibility of drug resistance. However, the major hurdle for miRNA-based therapies lies in the lack of a specific delivery system, a problem shared with all forms of gene therapy in cancer. In spite of the enormous amounts of biomarkers identified in epigenetic regulation of breast cancer and BCSCs, currently, there are only a few drugs available, and even less entering clinical trials. Therefore, in the future, the development of novel drugs or combination regimens are urgently required.

## Figures and Tables

**Figure 1 ijms-22-08113-f001:**
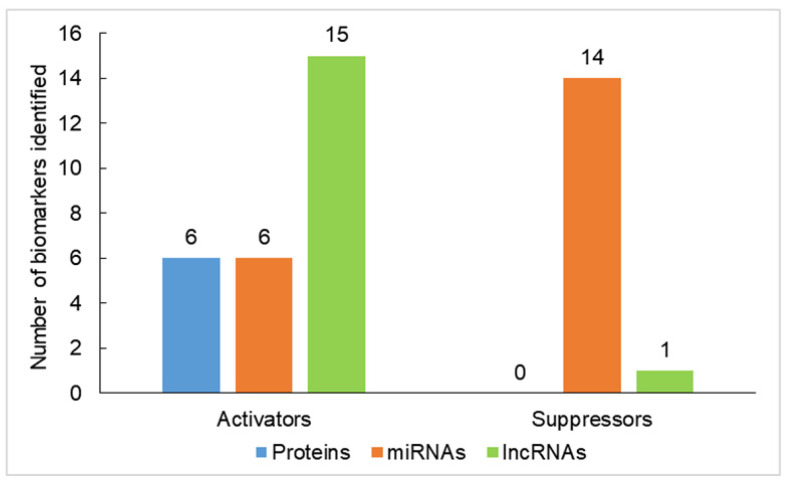
Histogram model showing the distribution of tumor activators and suppressors recently identified in epigenetic regulation of carcinogenesis for BCSC.

**Figure 2 ijms-22-08113-f002:**
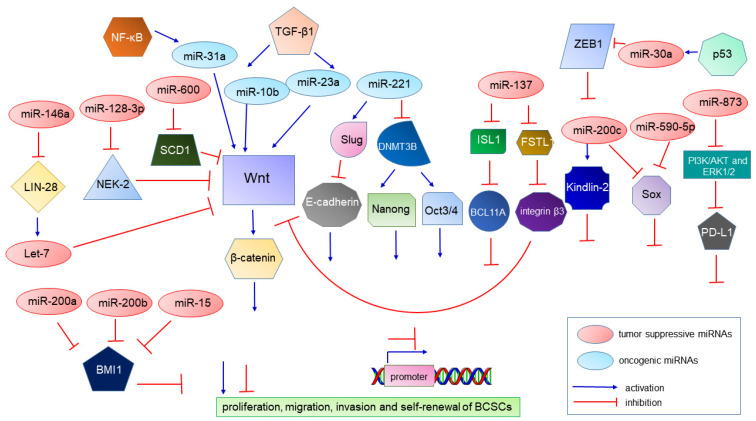
Schematic drawing representing the interplay between miRNA and protein involved in epigenetic regulation and tumorigenesis of BCSC.

**Table 1 ijms-22-08113-t001:** Biomarkers involved in epigenetic regulation of carcinogenesis for BCSCs.

Biomarkers	Function	Role in Carcinogenesis	References
Oncogenes or tumor activators			
Proteins			
DNMT1	Catalyzes hypermethylation of the cytosines and represses gene transcription.	DNMT1 is highly expressed in CSCs in mammospheres and tumorospheres, and *DNMT1* deletion suppresses mammary tumorigenesis. *ISL1* is hypermethylated and downregulated by DNMT1 in breast cancers and BCSCs.	[76,77,78]
		DNMT1 silencing reduced MSFE in triple-negative breast cancer (TNBC).	[79]
BMI1 (B lymphoma Mo-MLV insertion region homolog)	Component of PRC1 that plays a crucial role in epigenetic regulation of various physiological processes, such as cell differentiation, stem cell self-renewal and gene-silencing, through histone modifications.	Increases self-renewal capacity of BCSCs and promotes EMT.	[80,81,82]
EZH2	A PRC2 group protein and a HMT that methylates H3K27 and functions as a transcriptional repressor.	Inhibits the expression of tumor suppressor genes, such as P16 INK4a, E-cadherin, BRCA1 and the adrenergic receptor β2. Activates the Notch1 expression and signaling, leading to stem cell expansion in TNBC.	[83,84,85]
SETDB1 (SET Domain Bifurcated Histone Lysine Methyltransferase 1)	HMT catalyzes the di- and tri-methylation of H3K9 to induce gene-silencing.	Promotes breast cancer metastasis through the acquisition of stem-cell-like properties and EMT.	[86,87,88]
LSD1	Removes methyl groups from methylated proteins, including H3K4 and non-histone proteins, such as p53 and DNMT1.	Induces gene expression involved in EMT and elicits the BCSC program.	[89,90]
HDAC	Catalyzes the hydrolytic removal of acetyl groups from histone lysine residues.	Plays a significant role in the epigenetic regulation of CSC miRNAs.	[65,91]
miRNAs			
miR-10b	oncomiR	Contributes to TGF-β1-induced EMT and tumor metastasis.	[92,93]
miR-23a	oncomiR	Contributes to TGF-β1-induced EMT and tumor metastasis.	[94]
miR-221	oncomiR	Suppresses *CDH1* resulting in E-cadherin suppression. Represses the *DNMT3B* gene, and this leads to suppression of the *NANOG* and *OCT 3/4* genes and contributes to stemness maintenance in breast cancer.	[95,96]
miR-221/222 cluster	oncomiR	Induces the growth, migration, invasion and propagation of BCSCs.	[97,98]
miR-31	oncomiR	Increases the BCSC subpopulation and tumor initiation and metastasis abilities.	[99]
miR-520b	oncomiR	Is upregulated in breast cancer tissue and BCSCs and promotes the stemness.	[100]
lncRNAs			
HOTAIR	During embryonic development, HOTAIR regulates the silencing of the distant *HOXD* locus.	Downregulates miRNA-7 associated with EMT and STAT3 activity.	[101,102,103,104]
SOX2OT and linc00617	oncogenes	The stemness factor SOX2 is upregulated in BCSCs by SOX2OT and linc00617.	[105,106]
lncRNA-Hh	Oncogene	The self-renewal HH pathway is activated, which promotes CSC maintenance.	[107]
H19	Functions in the epigenetic silencing of the I*GF2* gene.	Promotes metastasis through EMT induction. Overexpression of H19 enhances clonogenicity, migration and mammosphere formation.	[108,109,110,111,112,113]
LncRNA-HAL	Oncogene	HAL silencing increases cell proliferation and impairs the proportion and function of CSCs.	[114]
LINC01133	Oncogene	Induces the BCSC phenotype.	[115]
lncRNA EPIC1	Oncogene	Overexpression of EPIC1 is correlated to poor survival outcomes in luminal B breast cancer.	[116]
lncRNA SOX21-AS1	Oncogene	Promotes BCSC properties and carcinogenesis via inhibiting Sox2 or the Hippo signaling pathway.	[117,118]
lncRNA THOR	Oncogene	Silencing of THOR induces reductions in mammosphere formation, stemness marker expression and ALDH1 activity of BCSCs.	[119]
LncCCAT1	Oncogene	Is significantly upregulated in breast cancer tissue and BCSCs, leading to poor patient outcomes.	[120]
LncRNA LUCAT1	Oncogene	Is expressed in breast cancer tissue and highly expressed in BCSCs, and regulates stemness features.	[121]
MALAT1	Has vital biological implications.	Is overexpressed in BCSC MCF7, and its knockdown decreases the proportion of BCSC MCF7 and mammosphere formation.	[110,122]
lncRNA FEZF1-AS	Oncogene	Knockdown of LncRNA FEZF1-AS reduces the ability of BCSC to form mammospheres, the expression of stem cell markers and the rate of CD44+/CD24− production.	[123]
lncRNA LINC00511	Oncogene	Is highly expressed in breast cancer, which is correlated with the poor prognosis of patients.	[124]
Tumor suppressors			
miRNAs			
miR-200c	A tumor suppressor	Targeting the self-renewal gene *Bmi-1* represses tumorigenicity of human BCSCs in vivo, and also targets Pin1 to regulate stemness of human primary BCSCs.	[125,126]
miR-200a, miR-200b and miR-15	Tumor suppressors	Overexpression of miR-200a, miR-200b and miR-15 decreased BMI1 and Ub-H2A protein expression in the CD44+ CSC population of MDA-MB-231 cells.	[127]
miR-200b	A tumor suppressor	Inactivates *FERMT2* and results in the inhibition of EMT and metastasis.	[128]
let-7	A tumor suppressor	Plays a significant role in BCSC, which is controlled via DNA methylation.	[129,130]
miR-30a	A tumor suppressor	Suppresses the *ZEB2* expression and controls aggressiveness.	[131]
miR-590-5p	A tumor suppressor	Inhibits stemness by targeting the *SOX2* gene and leads to a decrease in the BCSC population.	[132]
miR-140	A tumor suppressor	Inhibits stemness by targeting the *SOX2* gene and leads to a decrease in the BCSC population.	[133]
miR-146a	A tumor suppressor	Plays a role in mediating the induction and maintenance of BCSCs during EMT.	[134,135,136]
miR-600	A tumor suppressor	Reduces BCSC self-renewal through the inhibition of Wnt.	[137]
miR-128-3p	A tumor suppressor	Inhibits cell proliferation, migration, invasion and self-renewal of BCSCs.	[138]
miR-137	A tumor suppressor	Inhibits stemness and chemoresistance.	[139]
miR-873	A tumor suppressor	Inactivates PI3K/AKT and ERK1/2 signaling and attenuates the stemness and chemoresistance abilities of BCSCs.	[140]
lncRNAs			
lncRNA FGF13-AS1	A tumor suppressor	Is downregulated in breast cancer and inhibits glycolysis and stemness properties of breast cancer cells.	[141]

**Table 2 ijms-22-08113-t002:** The therapeutic implications targeting epigenetic regulation for breast cancer.

Targets	Mechanism of Treatment	Treatment	Status	References
DNMT	Lead to hypomethylation and gene de-repression, resulting in preventing EMT.	DNMT inhibitors: 5-AzaC (Vidaza) and 5-aza-20-deoxycytidine 5-AzaDC (decitabine)	The single use of 5-AzaDC or HDACi has been approved by the FDA for hematologic malignancies.	[200,201,202]
		Guadecitabine (SGI-110)	Phase 2 clinical trial	[202]
JMJD2	Attenuates the growth of breast cancer cells.	NCDM-32B, a JMJD2 inhibitor	Tested on the breast cancer cells	[203]
EZH2	Demonstrated a strong anti-cancer effect by inhibiting breast tumor growth and metastasis.	ZLD1039, an EZH2 inhibitor	Tested on xenograft-bearing mice	[204]
G9a	Induces apoptosis and impairs cell migration, cell cycle and anchorage-dependent growth in breast cancer cells.	BIX-01294, a G9a inhibitor	Tested on cells	[205]
HDAC	Induces PTEN membrane translocation through PTEN acetylation at K163 by inhibiting HDAC6, and this PTEN activation leads to inhibition of tumor growth.	HDACi, such as Trichostatin A (TSA) or suberoylanilide hydroxamic acid (SAHA)	Tested on xenograft tumor model	[206]
	Inhibits HDAC1 and HDAC7, that may therefore modify the epigenetic markers that characterize CSCs.	HDACi, TSA, a pan HDACi	Tested on the mouse model	[207]
	Inhibits HDAC3.	A pan-HDAC inhibitor (HDACi), AR-42	Tested on the mouse model	[208]
LSD1	Alter promoter activity of multiple genes in breast cancer cells.	LSD1 inhibitors: bizine, the tranylcypromine derivatives NCL1 and GSK2879552, biguanide, bisguanidine polyamine analogs and GSK287	Under clinical evaluation for cancer treatment	[209,210]
	Alter promoter activity of multiple genes in breast cancer cells.	polyamine analog inhibitors of LSD1, 2d or PG11144	Tested on cells	[211]
	Partially inhibited CSC formation.	LSD1 inhibitor pargyline	Tested on mouse model	[90]
	Reduced tumor growth of patient-derived CSCs.	LSD1 inhibitor QC6352	Tested on the xenograft model	[212]
	Induces significant growth arrest and apoptosis	A dual HDM inhibitor (MC3324)	Tested on both xenograft mice and chicken embryo models	[213]
	Could reduce colony formation and a decrease in SOX2 expression	LSD1 inhibitor iadademstat (ORY-100)	Tested on the patient-derived xenograft model	[214]
	Targeting BCSCs in vitro and in vivo	KDM1A inhibitor NCD38	Tested on in orthotopic xenograft models	[215]
miR-34	Epigenetic restoration of miR-34 could sensitize cancer cells to drugs and suppress stem cell features.	miR-34-based drug MRX34	Passed phase I clinical studies	[216]
miR-148a	Inhibits the BCSC properties via miR-148a-mediated inhibition of the TGF-β-SMAD2 signaling pathway.	CaA	Tested on mouse xenograft model	[217]
		glabridin	Tested on mouse xenograft models	[218]
let-7	The suppressive effects exerted by let-7 on stem-like cells involved let-7c/ER/Wnt signaling.	Tamoxifen (ER modulator)	Tested on xenografted tumor model	[219]
	Efficient liposomal delivery system for the combination of miRNA and siRNA to target the BCSCs.	Herceptin-conjugated cationic immuno-liposome with hyaluronic acid and protamine	Tested on cells	[220]
miR-26b	Suppresses BCSC metastasis via the miR-26b/YAF2 axis.	TV-circRGPD6 nanoparticle	Tested on orthotopic xenograft models	[221]
Combination therapy				
DNMT and HDAC	Reprogram aggressive TNBC cells that have undergone EMT into a less aggressive phenotype. Suppressed TNBC cell proliferation, colony formation, motility and stemness of the cancer cells in vitro and in vivo.	DNMTi, guadecitabine, in combination with HDACi, entinostat	XtMCF cells in CB17/SCID mice	[222]
DNMT and PARP	Enhanced tight binding of talazoparib to DNA and increased DSB formation and cytotoxicity.	DNMTi combined with PARPi, talazoparib	Tested on xenograft tumors	[223]
DNMT and PARP	DNMTi increased PARP trapping and reprogramed the DNA damage response to cause HRD, sensitizing BRCA-proficient cancer cells to PARPi.	DNMTi and PARPi	Will be tested in a phase I/II TNBC clinical trial	[224]
EZH2 and PARP	Combination showed increased sensitivity to PARP inhibition.	Combined PARP inhibition and EZH2 inhibition	Tested on xenograft model	[225]
LSD1 and HDAC	LSD1 interacts with HDACs to control breast cancer cell growth.	Combined treatment of LSD1 inhibitor, pargyline and HDACi, SAHA	Tested on cells	[226]
LSD1	Reduced stem cell potential and increased chemo-sensitivity.	LSD1 inhibitor combined with doxorubicin	Tested on xenograft model	[227]
LSD1 and PD-1	Suppressed tumor growth and pulmonary metastasis.	LSD1 inhibition in combination with anti-PD1 antibodies	Tested on xenograft tumors	[228]
lysine-specific histone demethylase-1A/HDAC, PARP and PD-1/PD-L1	Targeting BCSC and overcoming resistance and recurrence.	Lysine-specific histone demethylase-1A inhibitors or HDACi and PARPi/anti-PD-1/PD-L1	-	[229]
HDAC and PD-L1	Inhibits tumor growth and increases survival.	HDACi and immune checkpoint inhibitors	Tested on breast cancer mouse model	[230]
HDAC and JAK/BRD4	JAK/BRD4 inhibition sensitizes TNBC cells to HDAC inhibitions.	The combination of HDACi and JAK/BRD4 inhibitors	Tested on mouse models	[231]
HDAC	The objective response rates were comparable to those achieved with the previously approved ixabepilon monotherapy or combination with capecitabine.	Combination of the cytostatic drug ixabepilon with HDACi vorinostat	A phase Ib study	[232]
HDAC	Enhanced toxicity and increasing autophage induction.	The combination therapy of HDACi YCW1 with ionizing radiation. This led to induction of cell death in TBNC cell lines in vitro and in mouse models.	Tested on orthotopic mouse model	[233]

## Data Availability

Not applicable.

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
