# Peer review of "Epigenetic Regulation of Breast Cancer Stem Cells Contributing to Carcinogenesis and Therapeutic Implications"

_ijms, 2021, doi:10.3390/ijms22158113_

Round 1

Reviewer 2 Report

Comments

  1. Please change the abbreviation in the abstract form line 25 to 19 (BCSCs)
  2. In 2021, 281,550 new cases of breast cancer to be diagnosed in women- line 31 change the wording
  3. Line 80-82 please check the sentence once, there is a mismatch in the sentence
  4. Line 85- “such as loe expression” I think it is “low”
  5. So many places the authors are talked about ncRNA and lncRNA, please write the full form and use short forms when it is representing first time
  6. Please modify the Table 1 where the alignment is not proper and some places the wordings are missing
  7. Abbreviate few headings and few words like DNMT1, TET1, BMI1, EZH2, SETDB1, LSD1 etc…
  8. Authors can make the table 1 into the repressor and suppressors  for the Biomarkers involved in epigenetic regulation of carcinogenesis for BCSC
  9. The authors can represent the biomarkers which shows the repressor and suppressors  in epigenetic regulation of carcinogenesis for BCSC in histogram model or another representation model
  10. Please check all the abbreviations are elaborated all over the manuscript

Overall the manuscript is written well and the literature is up to date but the authors should more focus on the mechanism that helps for to do the more interested research on the present topic.
